

# 1 Trace gas fluxes from tidal salt marsh soils: implications for carbon-
# 2 sulfur biogeochemistry

Margaret Capooci[1] and Rodrigo Vargas [1]
[1] Department of Plant and Soil Science
University of Delaware
152 Townsend Hall
531 South College Ave.
Newark, DE, USA 19716
*Correspondence to:*
Rodrigo Vargas
152 Townsend Hall
531 South College Ave.,
Newark, DE 19716
rvargas@udel.edu
Phone: 302-831-1386



**Abstract**
Tidal salt marsh soils can be a dynamic source of greenhouse gases such as carbon dioxide
($CO_2$), methane ($CH_4$), and nitrous oxide ($N_2O$), as well as sulfur-based trace gases such as
carbon disulfide ($CS_2$) and dimethylsulfide (DMS) which play roles in global climate and
carbon-sulfur biogeochemistry. Due to the difficulty in measuring trace gases in coastal
ecosystems (e.g., flooding, salinity), our current understanding is based on snap-shot
instantaneous measurements (e.g., performed during daytime low tide) which complicates our
ability to assess the role of these ecosystems for natural climate solutions. We performed
continuous, automated measurements of soil trace gas fluxes throughout the growing season to
obtain high-temporal frequency data and to provide insights into magnitudes and temporal
variability across rapidly changing conditions such as tidal cycles. We found that soil $CO_2$ fluxes
did not show a consistent diel pattern, $CH_4$, $N_2O$, and $CS_2$ fluxes were highly variable with
frequent pulse emissions (>2,500%, >10,000%, and >4,500% change, respectively), and DMS
fluxes only occurred mid-day with changes >185,000%. When we compared continuous
measurements with discrete temporal measurements (during daytime, at low tide), discrete
measurements of soil $CO_2$ fluxes were comparable with those from continuous measurements,
but misrepresent the temporal variability and magnitudes of $CH_4$, $N_2O$, DMS, and $CS_2$.
Discrepancies between the continuous and discrete measurement data result in differences for
calculating the sustained global warming potential (SGWP), mainly by an overestimation of $CH_4$
fluxes when using discrete measurements. The high temporal variability of trace gas fluxes
complicates the accurate calculation of budgets for use in blue carbon accounting and earth
system models.



## 1. Introduction

Coastal vegetated ecosystems such as tidal salt marshes, mangrove forests, and seagrass beds provide a wide range of ecosystem services, such as mitigating storm surge and providing nursery areas for fish species (Barbier et al., 2011; Möller et al., 2014). They also store large amounts of carbon at rates forty times higher than tropical rainforests (Rosentreter et al., 2018; Duarte et al., 2005) and are referred to as "blue carbon" ecosystems. The importance of coastal vegetated ecosystems in climate change policies has been recognized by the Paris Agreement (UNFCCC, 2015). Prior to the Paris Agreement, there has been increased interest in better quantifying the net balance between carbon storage and carbon release in coastal vegetated ecosystems for both scientific and carbon market purposes. For example, the Verified Carbon Standard developed a methodology to assess and verify the amount of carbon removed from the atmosphere in tidal wetland and seagrass restoration projects for carbon market purposes (Emmer et al., 2021). However, there are major knowledge gaps in assessing blue carbon in coastal vegetated ecosystems. Specifically, the high spatial and temporal variability of greenhouse gas (GHG) emissions, particularly for $CH_4$ and $N_2O$, in coastal vegetated ecosystems complicates blue carbon offset calculations (Rosentreter et al., 2021; Capooci et al., 2019; Al-Haj and Fulweiler, 2020; Murray et al., 2015). Thus, there is a need for developing measurement protocols to fully quantify the contribution of multiple GHGs in blue carbon ecosystems.

To improve our understanding of blue carbon ecosystems in global biogeochemical cycles we need to think beyond traditional GHG trace gases (i.e., $CO_2$, $CH_4$, $N_2O$). Tidal salt marshes produce sulfur-based trace gases due to the prevalence of sulfur cycling within their soils, which has implications for carbon-sulfur biogeochemistry and the global climate. While coastal areas are major sources of sulfur gases (Kellogg et al., 1972), there is large



uncertainty in emission rates (Carroll et al., 1986; Andreae and Jaeschke, 1992). Dimethyl
sulfide (DMS) is one of the dominant sulfur-based gases emitted from salt marshes (Hines,
1996), and dimethylsulfoniopropionate (DMSP), a DMS precursor, can be produced by salt
marsh plant species *Spartina alterniflora*, *S. anglica*, and *S. foliosa* (Hines, 1996). DMS plays an
important role in linking together carbon and sulfur biogeochemistry in salt marsh soils. It can be
decomposed by not only sulfate-reducing bacteria, but can also act as a non-competitive
substrate for methylotrophic methanogenesis (Kiene, 1988; Kiene and Visscher, 1987; Oremland
et al., 1982) which allows methane production to occur in soils dominated by sulfate reduction
(Seyfferth et al., 2020). Another sulfur-based trace gas released from tidal salt marshes is carbon
disulfide ($CS_2$). $CS_2$ can be produced by biological processes (Brimblecombe, 2014) and is a
precursor to carbonyl sulfide (COS; Whelan et al., 2013). COS is the most abundant reduced
sulfur compound in the atmosphere and can form sulfate aerosols that affect the Earth's radiative
properties by reflecting sunlight, thereby having a cooling effect on the climate (Watts, 2000;
Taubman and Kasting, 1995). Despite sulfur-based trace gases playing a role in wetland soil
biogeochemistry and in global climate, there is a need to quantify coastal wetland sulfur
emissions and to connect those emissions to both the salt marsh sulfur cycle and to global
budgets (DeLaune et al., 2002; Whelan et al., 2013).

Historically, both soil GHGs and S-based fluxes are measured using manual survey

chambers, particularly during daytime low tide (e.g., De Mello et al., 1987) when soils are less
likely to be submerged and are accessible to researchers. Manual measurements have a number
of advantages, including the ability to sample over large areas over short periods of time
(Moseman-Valtierra et al., 2016; Simpson et al., 2019), but these measurements are labor-
intensive and provide limited information regarding temporal variability (Koskinen et al., 2014;



Savage et al., 2014; Vargas et al., 2011). On the other hand, recent advances in high temporal-
frequency soil efflux measurements (Capooci and Vargas, 2022; Diefenderfer et al., 2018;
Järveoja et al., 2018) have provided researchers with unprecedented temporal information to
better understand diel and tidal patterns, as well as the influence of pulse events on trace gas
emissions within salt marshes. While the use of automated systems is becoming more common
in measuring salt marsh fluxes, their use is limited by high instrumentation costs, electricity
requirements, and logistical challenges associated with installing these instruments in an
environment prone to flooding and with high humidity. As automated systems become more
prevalent, it provides researchers with the opportunity to evaluate data collected from manual
measurements, such as daily means, that have been used to inform models and budgets,
particularly for understudied trace gases such as $N_2O$, $CS_2$, and DMS.

The objective of this study is to characterize the spatial and temporal variability of trace

gases from soils in a tidal salt marsh. Specifically, we focus on $CO_2$, $CH_4$, $N_2O$, $CS_2$, and DMS
to assess the differences between measurements taken at a particular time of day (i.e., daytime
low tide) and measurements with high-temporal frequency (i.e., continuous measurements). Few
studies have measured GHG fluxes from tidal salt marshes using continuous, automated
measurements (Diefenderfer et al., 2018; Capooci and Vargas, 2022), and this is a pioneering
study that provides unprecedented information about the magnitudes and patterns of $CS_2$ and
DMS fluxes via continuous measurements. Furthermore, this study tests whether traditional
measurement protocols based on discrete temporal measurements provide similar information as
data derived from continuous measurements, including the calculation of the sustained global
warming potential (SGWP). Development of new technologies and incorporation of this



information has important implications for calculating greenhouse and trace gas budgets, as well
as the role salt marshes play in global biogeochemical cycles.

**2. Materials and methods**

*2.1 Study site*

The study was conducted at St. Jones Reserve, the brackish estuarine component of the

Delaware National Estuarine Research Reserve. The site is part of the Delaware Estuary and is
tidally connected to the Delaware Bay via the St. Jones River. St. Jones is classified as a
mesohaline tidal salt marsh (DNREC, 1999) and has silty clay loam soils (10% sand, 61% silt,
29% loam, Capooci et al 2019). The study was conducted in a section of the marsh dominated by
*Spartina alterniflora* (= *Sporobolus alterniflorus* (Loisel.); Peterson et al., 2014) and will be
referred to as SS as established in previous studies (Seyfferth et al., 2020; Capooci and Vargas,
2022). This area is lower in elevation relative to the rest of the marsh, is characterized by sulfur
reduction (Seyfferth et al., 2020), and covers ~66% of the salt marsh landscape (Vázquez-Lule
and Vargas, 2021).

*2.2 Experimental set-up*

The experiment was performed over the course of 6 campaigns to cover a full growing

season: greenup (G), maturity (M), senescence (S), and dormancy (D) as described by the
canopy phenology of the study site (Hill et al., 2021). The campaigns began during the latter half
of the 2020 growing season and continued into beginning of the 2021 growing season season
(M1 – 29 June to 2 July, M2 – 31 July to 3 Aug , S1 – 31 Aug to 3 Sept, S2 – 28 Sept to 1 Oct,



D1 – 13 Apr to 16 Apr, and G1 – 31 May to 3 June) due to delays related to the COVID-19
pandemic. We installed six PVC collars (diameter: 20 cm), placed ~1.2 meters apart, four
months prior to the beginning of the experiment in the year 2020. Any vegetation that grew
inside these collars in between campaigns was carefully removed prior to the start of the
measurements. These collars were used to set down six automated chambers (LICOR 8100-104,
Lincoln, Nebraska) to measure trace gas fluxes as described below.

*2.3 Trace gas flux measurements and QA/QC*
The autochambers were coupled with a closed-path infrared gas analyzer (LI-8100A,
LICOR, Lincoln, Nebraska) and a Fourier transform infrared spectrometer (DX4040, Gasmet
Technologies Oy, Vantaa, Finland). The LI-8100A and the DX4040 were connected in parallel
since the DX4040 has its own internal pump and flow rates. Trace gas fluxes were measured
once per hour per chamber (i.e., all six chambers were measured within an hour). Measurements
were 5 minutes long and each chamber was flushed for 5 minutes total (pre-purge and post-purge
were both 2.5 minutes long) to help reduce the impacts of humidity on the instruments. Each
campaign lasted approximately 72 hours where approximately 416 measurements were recorded.
At the beginning of each campaign and every 24 hours after, we performed a zero
calibration on the DX4040 using ultra-pure 99.999% $N_2$ gas. It is recommended that zero
calibrations are performed every 24 hours and when the ambient temperature changes by 10°C,
so the experiment was paused for ~30 minutes during the zero calibrations each day. Gas fluxes
were calculated using Soil Flux Pro (v4.2.1, LICOR, Lincoln, Nebraska) and underwent
standardized quality assurance and quality control protocol as established in previous
publications (Capooci et al., 2019; Petrakis et al., 2017). Briefly, QAQC included removing all





values due to instrumental errors, comparing exponential and linear fits to select for the
measurement with the higher $R^2$, removing all measurements during times where the $R^2$ for $CO_2$
< 0.90, and removing all negative $CO_2$ fluxes.

*2.4 Ancillary measurements*
Meteorological (station: delsjmet-p) and water quality (station: Aspen Landing) data were
obtained from the National Estuarine Research Reserve's Centralized Data Management Office
(CDMO) and collected according to their protocol (System-wide Monitoring Program).
Meteorological data was collected using a CR1000 Meteorological Monitoring Station
(Campbell Scientific, Logan, UT, USA). Water quality data were measured using a YSI 6600
sonde (YSI Inc., OH, USA). Both data sets were cleaned and gap-filled following the protocol
established in Capooci et. al. (2022).
Phenological data were obtained from the PhenoCam network (site: stjones,
Seyednasrollah et al., 2019) as described previously (Trifunovic et al., 2020; Hill et al., 2021).
Briefly, a single mid-day photo (12:00:00 h) was selected for each of the days in the study period
and was visually inspected to remove images with obvious distortions. Since the images included
a variety of vegetation types, the region of interest delineated to only the area containing *S.*
*alterniflora*, the main species at the study site. Then the phenopix R package (Filippa et al.,
2020) was used to extract and calculate the greenness index, as well as delineate the phenophases
for the study period (Hill et al., 2021).

*2.5 Data analyses*


Daily averages and associated standard deviations were calculated for meteorological and
water quality data, except for the greenness index. Soil trace flux data were averaged into hourly
and daily means and standard deviations. For heat maps, average hourly and campaign-length
coefficients of variation were calculated.
We extracted measurements from the time series of the automated measurements to
represent information collected from discrete temporal measurements conducted during daytime
low tide. This approach aimed to represent a measurement protocol derived from manual (i.e.,
survey) measurements where most measurements are performed at daytime and low tide  for
logistical reasons. To identify and extract these measurements, we identified when low tide
occurred during each day (between 9:00:00 and 17:00:00 h) of the campaigns from water level
data obtained from the tidal creek. All automated measurements that fell between 1 hour before
and 1 hour after low tide were extracted, averaged into a daily value, and classified as "discrete"
measurements. For example, if low tide fell at 13:00:00 h, all continuous measurements that fell
between 12:00:00 and 14:00:00 h were then extracted and averaged to obtain a daily mean. Daily
means were also calculated for all automated measurements collected during the day and will be
referred to as the "continuous" daily mean. Differences in the means and distributions of the
continuous and discrete fluxes were assessed using a t-test and a Kolmogorov-Smirnov test,
respectively.
Sustained global warming potential (SGWP) was calculated for both the campaign-long
and daytime low tide fluxes for $CO_2$, $CH_4$, and $N_2O$. SGWP accounts for sustained gas emissions
over time compared to the global warming potential which accounts for a pulse emission over
time (Neubauer and Megonigal, 2019). To calculate the SGWP, data from Day 2 and 3 of each
campaign was used since measurements on Day 1 and 4 did not always occur during daytime





low tide. Fluxes were converted into g m$^{-2}$ and multiplied by the 20 and 100-year SGWP
(Neubauer and Megonigal, 2019). SGWP were compared to see whether extrapolating SGWP
from daily-averaged manual measurements done at low tide yielded similar values as hourly-
averaged from high temporal frequency measurements.

**3. Results**

*3.1 Meteorological and water quality*

Air temperature and greenness index show traditional seasonal patterns of temperate salt

marshes (Fig. 1). Daily mean air temperature ranged from -3.5℃ to 29.9℃, with an average
daily temperature of 13.8 ± 9.1℃, while greenness index ranged from 0.30 to 0.42 with an
average of 0.34 ± 0.04. Relative humidity, barometric pressure, water level, and salinity varied
throughout the year. Relative humidity ranged from 32.6% to 100% with an average of 79.1% ±
16.7%. Barometric pressure was between 999.7 and 1036 mb with an average value of 1018.3 ±
6.8 mb. Daily water level ranged from -0.30 m to 0.76 m with an average height of 0.25 ± 0.2 m,
while salinity ranged from 1.1 ppt to 20.4 ppt with an average of 8.0 ± 4.45 ppt.

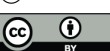

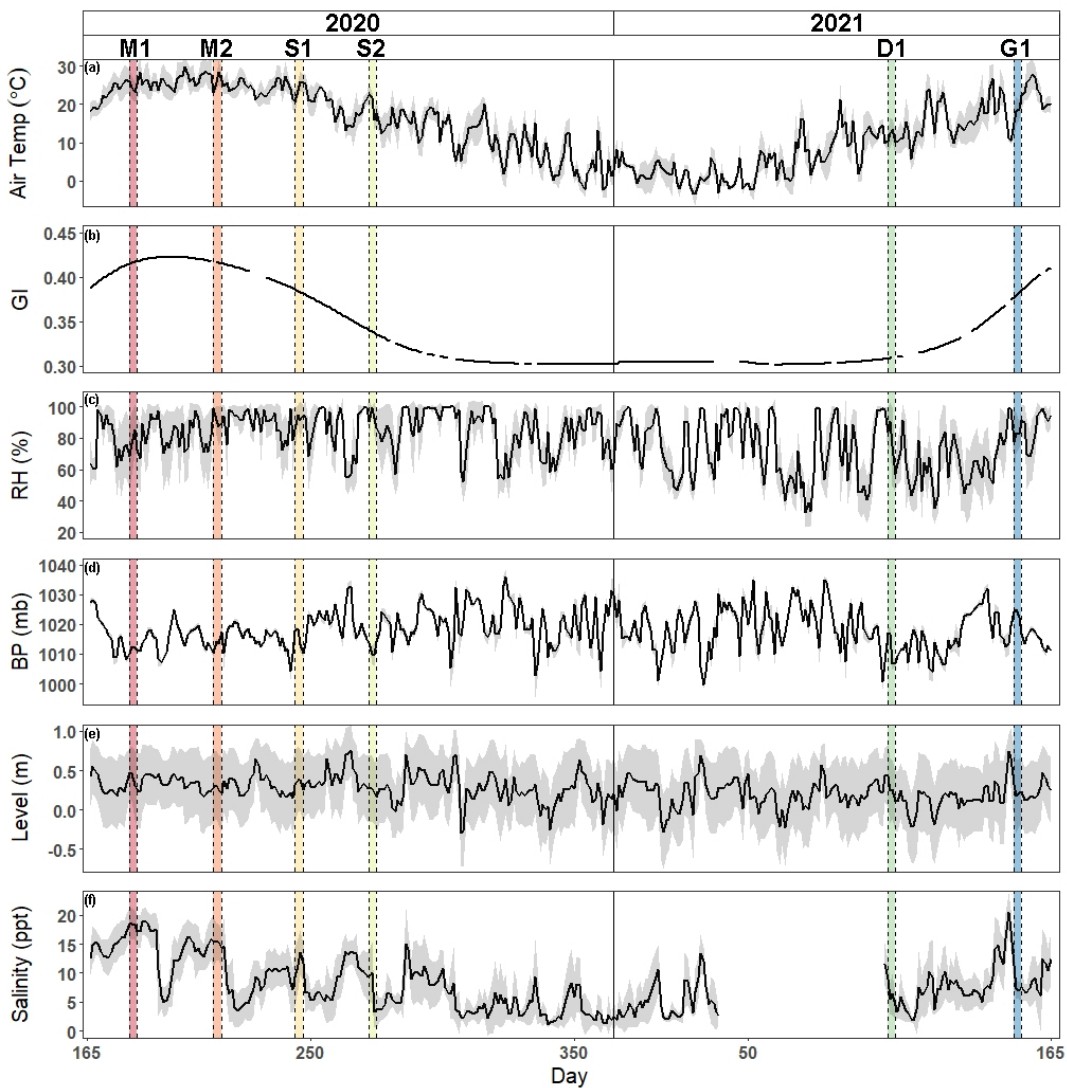


**Figure 1.** Time series of hourly mean ± SD (gray shaded region of (a) air temperature, (b)

greenness index, (c) relative humidity, (d) barometric pressure, (e) water level, and (f) salinity

from June 14, 2020 to June 14, 2021. Vertical shaded areas correspond to each of the campaigns

(M = maturity, S = senescence, D = dormancy, G = greenup).





*3.2 Greenhouse gas and sulfur-based trace gas patterns and variability*


Average $CO_2$ fluxes were significantly different in each campaign, with the highest
average fluxes occurring during the G1 campaign and the lowest during the D1 campaign (Fig.
2a). During some campaigns, such as S1, $CO_2$ fluxes did not show similar temporal patterns
between chambers, whereas during other campaigns, such as M2 and G1, all six chambers had
similar patterns. While there is a seasonal pattern in $CO_2$ fluxes, with higher fluxes occurring
during warmer months, diel patterns were not consistent between campaigns. One notable
exception is the G1 campaign, during which a clear diel pattern was observed. $CO_2$ fluxes had
consistent variability from one hour to the next during each of the 6 campaigns (Fig. 3a), with
overall average variability ranging from 28.9% during M2 to 49.6% during Dl.
$CH_4$ fluxes were low most of the time, particularly during the G1 campaign (Fig. 2b).
However, $CH_4$ pulses occurred during 5 out of the 6 campaigns, with S1and S2 having the most
frequent pulse emissions. S2 had the largest $CH_4$ pulse,13,488 nmol m$^{-2}$ s$^{-1}$, which was 2,599%
higher than the average flux. The highest average $CH_4$ fluxes also occurred during S1 and S2,
while the highest hourly variability occurred in both S1 and S2, as well as in M2 (Fig. 3b). Mean
$CH_4$ variability ranged from -108% in M1 to 91.0% in S1.
Most $N_2O$ fluxes were near-zero, with periodic pulses of emissions or uptake that ranged
from -33.8 to 19.0 nmol m$^{-2}$ s$^{-1}$ (Fig. 2c), with a maximum percent change from the mean of
10,231%. Four out of the six campaigns (M1, S2, D1, and G1) had net $N_2O$ uptake, while two
campaigns (M2, S1) had net $N_2O$ fluxes. There were no significant differences between
campaigns except for M1 and S1. Meanwhile, $N_2O$ fluxes had very high hourly variability
ranging from -106,964% to 26,208% (Fig. 3c). Consequently, average variability during each
campaign was highly variable from -1,032% to 129%.



Similarly to $CH_4$ and $N_2O$, $CS_2$ fluxes were low the majority of the time, with occasional
pulses of emissions or uptake (Fig. 2d). $CS_2$ fluxes ranged from -386.9 to 306.2 nmol m$^{-2}$ s$^{-1}$,
with a maximum percent change from the mean of 4,785%. All campaigns had net emissions
despite periodic pulses of $CS_2$ uptake. $CS_2$ fluxes also had high hourly variability, with overall
means for each campaign ranging from -70.2% during D1 to 2254% during M2 (Fig. 3d).
DMS emissions were zero for most of the campaigns (Fig. 2e). Pulses of emissions and
uptake tended to occur during mid-day. DMS fluxes ranged from -158.5 to 230 nmol m$^{-2}$ s$^{-1}$,
with a maximum percent change from the mean of 185,987%. D1 and G1 had net uptake, while
the other four campaigns had net emissions of DMS. During periods of emissions and uptake,
hourly variability ranged from -870.5% to 888.7% (Fig. 2e). The extended periods of no DMS
fluxes contributed to low overall mean variability during each campaign, ranging from -2.45% in
S2 to 35.7% in M2.



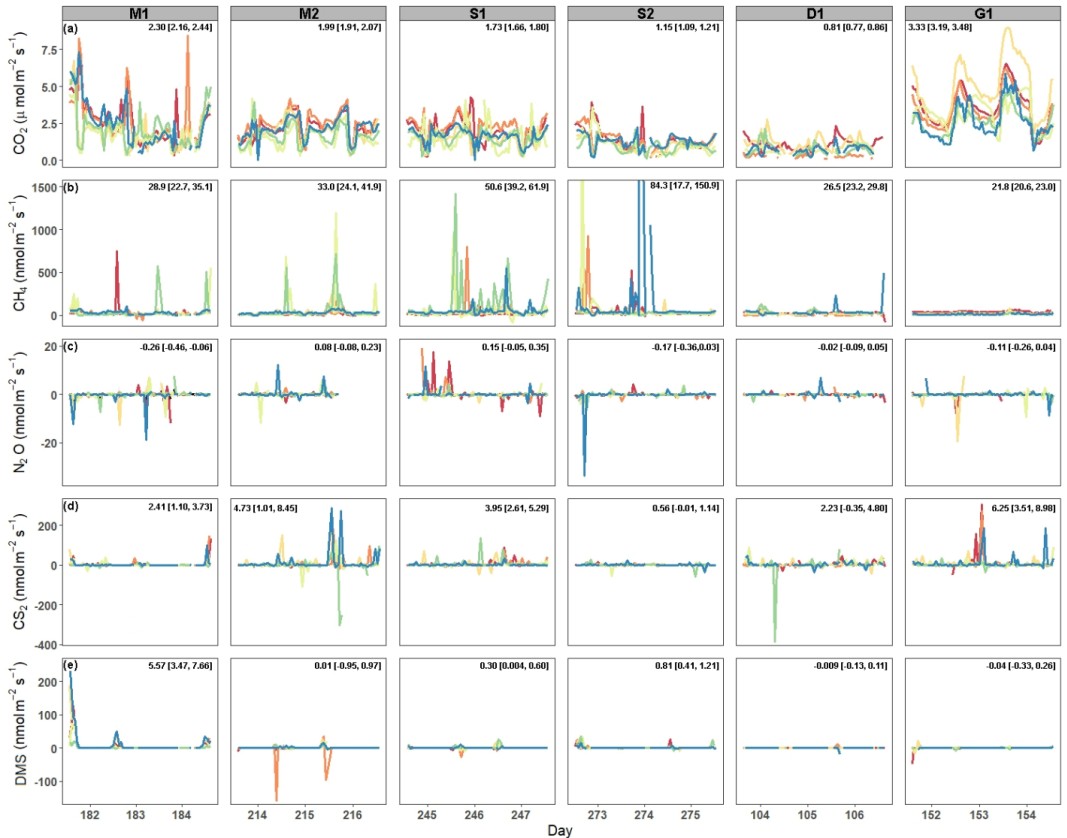


**Figure 2.** Time series of fluxes from each chamber during each campaign for (a) $CO_2$, (b) $CH_4$,

(c) $N_2O$, (d) $CS_2$, and (e) DMS. Each color designates a different chamber. The campaign means

[LCI, UCI] are listed on each panel. The y-axis for $CH_4$ fluxes was shortened to show the

variability. Full range of $CH_4$ fluxes during S2 can be seen in Supplementary Figure (SF) 1.


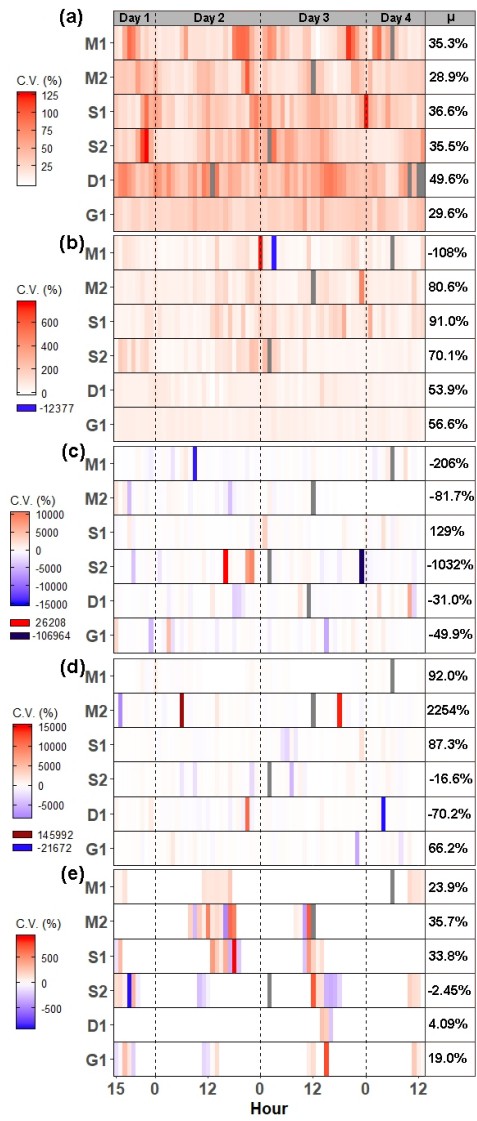

**Figure 3.** Heat maps of hourly coefficient of variance (CV) for (a) $CO_2$, (b) $CH_4$, (c) $N_2O$, (d)

$CS_2$, and (e) DMS during each campaign. Each pixel represents the average CV for that hour.

Mean CV for each campaign is listed in the μ column. Grayed out pixels represent NA. Note:

legend scale is different for each gas and campaigns start at 15:00:00 h on Day 1 and end at

13:00:00 h on Day 4.



*3.3 Comparisons between continuous and discrete measurement scenarios*


A subset of the continuous measurements that fall during daytime low tide was selected

to represent data collected using traditional discrete, manual measurements which are commonly
reported for tidal salt marshes. Information from continuous and discrete datasets are compared
to elevate whether they provide similar distributions, daily means, flux-temperature
relationships, and SGWP.

Continuous and discrete flux distributions can be seen via density plots (Fig. 4). While

the distributions for continuous and discrete fluxes overlap for each of the five gases, four of the
five gases have significantly different distributions of fluxes when comparing the continuous and
the discrete datasets (Table 1). The only gas that had similar distributions between the two
sampling intervals was $CO_2$ (Table 1). For all gases, the continuous distribution had higher
kurtosis values and higher C.V. than the discrete fluxes (Table 1). Of the five gases, $CS_2$ was the
only one with a more skewed discrete data distribution and significantly different means between
continuous and discrete measurement scenarios (Fig. 4b, Table 1).

For $CS_2$ and DMS, discrete measurements had higher overall daily mean fluxes (Fig. 5d,

5e), while the opposite occurred for $CH_4$ and $N_2O$ (Fig. 5b, 5c). $CO_2$ fluxes from continuous and
discrete measurements had nearly a 1:1 relationship (Fig. 5a). Both $CO_2$ and DMS had strong
relationships between continuous and discrete daily means, with r-squares higher than 0.7, while
$N_2O$ and $CS_2$ had moderate relationships. $CH_4$ had a poor fit between continuous and discrete
measurements.

Next, relationships between trace gas flux and air temperature were evaluated for each

gas under continuous and discrete measurement scenarios. $CO_2$ and $CH_4$ fluxes had statistically
significant relationships for both discrete and continuous measurements versus air temperature



(Fig. 6a-d).  Air temperature explained 38% and 21% of the variability for discrete and
continuous measurements for $CO_2$, respectively (Fig. 6a, b), while air temperature explained
32% and 7% of the variability for discrete and continuous measurement for $CH_4$ (Figs. 6c, d).
The slopes for both discrete and continuous $CO_2$ fluxes were not significantly different (95% CI;
0.029 - 0.12, 0.037 - 0.054, respectively), as well as for $CH_4$ (95% CI; 2.14 - 12.7, 1.31 - 2.71,
respectively). For $N_2O$, $CS_2$, and DMS, there were no significant relationships between discrete
daily mean fluxes and air temperature, but there were significant relationships between
continuous hourly mean fluxes and air temperature (Fig. 6e-j). Air temperature explained very
little variability for $N_2O$, $CS_2$, and DMS.

Discrete measurements had a higher SGWP potential than the continuous measurements.

While the discrete measurements had a slightly lower SGWP for $CO_2$ and a slightly higher
SWGP for $N_2O$, the difference between continuous and discrete SGWP was driven by $CH_4$. The
20-yr and 100-yr SGWP for discrete measurements of $CH_4$ were up to ~38% higher than the
respective continuous measurements, contributing to an overall increase of ~18% and ~11% for
the discrete measurement's 20- and 100-year SGWP.



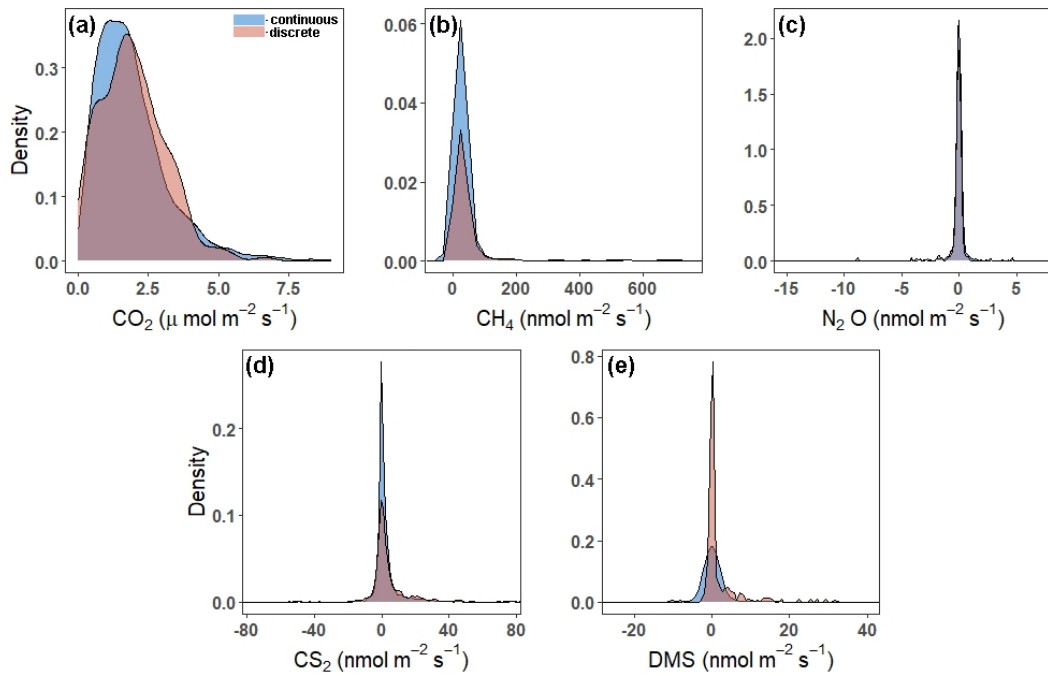


**Figure 4.** Density plots comparing the distribution of fluxes throughout all campaigns

(continuous) to those measured during daytime low tide (discrete) for (a) $CO_2$, (b) $CH_4$, (c) $N_2O$,

(d) $CS_2$, and (e) DMS. Note: The scales on the x- and y-axes are different. The tails have been

cut off to better see the peaks for (b), (c), (d), and (e). To see plots with full distributions, see SF

2.





**Table 1.** Summary of continuous and discrete measurement data and distributions for each gas.
An alpha of < 0.05 was used to determine significant differences between the means and the
distributions. Note: Means for $CO_2$ are in $\mu mol\ m^{-2}\ s^{-1}$, while the other gases are in $nmol\ m^{-2}\ s^{-1}$.

| Gas | Sampling Frequency | Mean | 95% CI | C.V. | Skewness | Kurtosis | Means Different? | Distributions Different? |
|---|---|---|---|---|---|---|---|---|
| $CO_2$ | Continuous | 1.92 | 1.86–1.97 | 67.2% | 1.53 | 6.51 | No | No |
| | Discrete | 1.90 | 1.74–2.07 | 62.3% | 0.67 | 3.65 | | |
| $CH_4$ | Continuous | 41.2 | 29.5–52.9 | 708% | 41.6 | 1903 | No | Yes p = 0.02 |
| | Discrete | 57.6 | 39.2–76.0 | 234% | 5.21 | 34 | | |
| $N_2O$ | Continuous | -0.06 | -0.13–0.009 | 2686% | -4.67 | 133 | No | Yes p < 0.001 |
| | Discrete | -0.16 | -0.29–-0.04 | 556% | -4.39 | 47.8 | | |
| $CS_2$ | Continuous | 3.39 | 2.45–4.33 | 673% | 1.51 | 116 | Yes p = 0.04 | Yes p = 0.05 |
| | Discrete | 6.44 | 3.70–9.18 | 312% | 3.93 | 22.9 | | |
| $DMS$ | Continuous | 1.11 | 0.70–1.51 | 907% | 8.74 | 223 | No | Yes p < 0.001 |
| | Discrete | 1.77 | 1.06–2.48 | 295% | 3.40 | 16.6 | | |


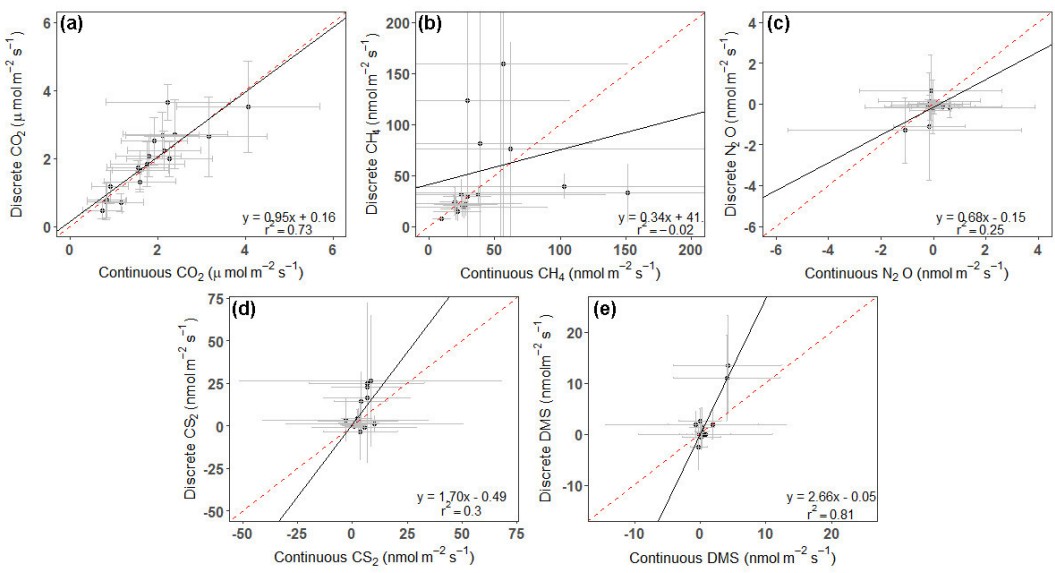



**Figure 5.** Plots comparing the daily average of continuous to discrete measurements for (a) $CO_2$,

(b) $CH_4$, (c) $N_2O$, (d) $CS_2$, and (e) DMS. Error bars represent the SD and have been cut off in

panel (b) to show data better. See SF 3 for full error bars for panel b. Red dashed line is the 1:1

line, while the black solid line is the trend line.

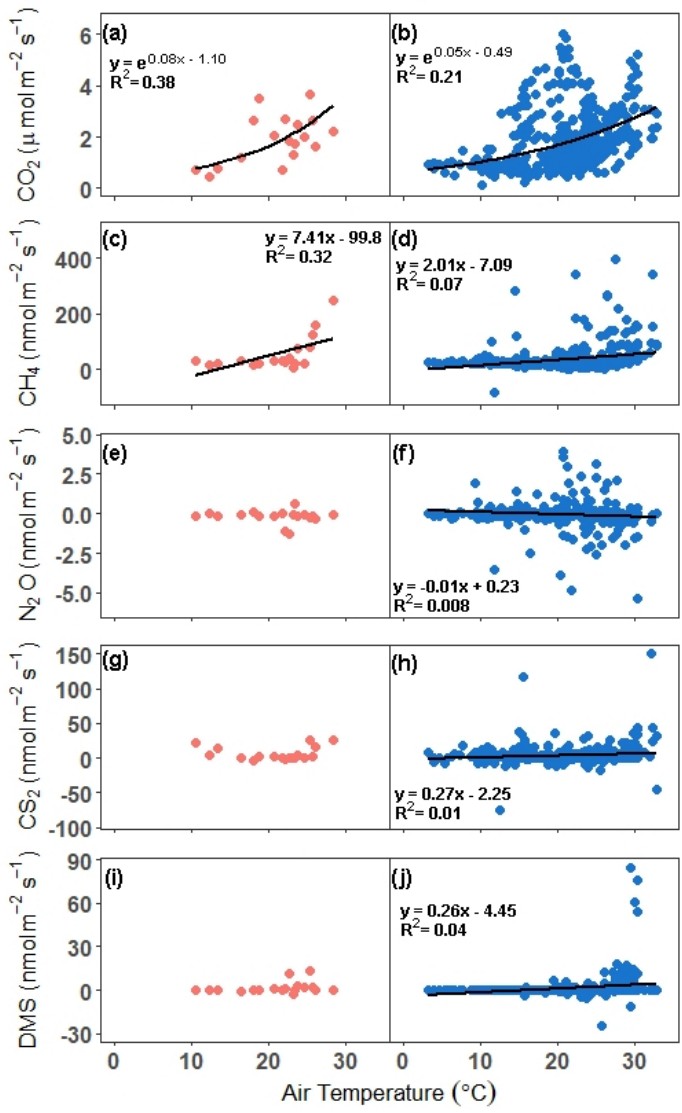

**Figure 6.** Comparison of fluxes versus air temperature for all campaigns. In panels, a, c, e, g, and

I, the hourly continuous mean is compared to the hourly air temperature, while in panels b, d, f,

h, and k, the discrete daily mean is compared to the daily air temperature. The trend lines for

significant relationships at alpha <0.05 are plotted. Note: In panel d, the outlier hourly mean of

2,275 nmol m$^{-2}$ s$^{-1}$ is not included in the trend line or in the graph.



**Table 2.** Sustained global warming potential (SGWP) derived from continuous and discrete temporal (during daytime low tide) measurements in a tidal salt marsh.

| Frequency | CO$_2$ (g m$^{-2}$) | CH$_4$ (CO$_{2\text{-eq}}$ (g m$^{-2}$)) | | N$_2$O (CO$_{2\text{-eq}}$ (g m$^{-2}$)) | | Total (CO$_{2\text{-eq}}$ (g m$^{-2}$)) | |
|---|---|---|---|---|---|---|---|
| | | 20-yr SGWP | 100-yr SGWP | 20-yr SGWP | 100-yr SGWP | 20-yr SGWP | 100-yr SGWP |
| Continuous | 84.9 | 70.4 | 33.0 | 0.27 | 0.30 | 155.57 | 118.2 |
| Discrete | 82.7 | 103.2 | 48.4 | 0.40 | 0.44 | 186.3 | 131.54 |

## 4. Discussion

*4.1 Measuring all the time: seasonal and diel patterns and hot moments of soil trace gases*

Spatial variability between the individual chambers at SS were low, but CO$_2$ fluxes showed temporal variability that corresponded to changes in temperature. The relatively low spatial variability within our experimental setting contrasts with previously reported high spatial variability of CO$_2$ fluxes attributed to the presence of a hot spot (Capooci and Vargas, 2022). However, previous CO$_2$ fluxes measured at the SS site ranged from 0-10 μmol m$^{-2}$ s$^{-1}$, with the bulk of the measurements between 0-4 μmol m$^{-2}$ s$^{-1}$, with higher fluxes associated with hot spots or warmer temperatures (Capooci and Vargas, 2022; Seyfferth et al., 2020). Therefore, location of measurements within a landscape could be influenced by hot spots, which complicates ecosystem scale calculations of soil CO$_2$ fluxes (Barba et al., 2018). In addition, there was a seasonal pattern evident in the CO$_2$ fluxes, with higher emissions during the growing season, as typical in temperate ecosystems, as well in the significant relationship between CO$_2$ and air temperature. Other studies at temperate wetland sites have found higher fluxes during the





summer (Simpson et al., 2019; Yu et al., 2019; Bridgham and Richardson, 1992), as well as
relationships between $CO_2$ fluxes and temperature (Capooci and Vargas, 2022; Simpson et al.,
2019; Xie et al., 2014) highlighting that $CO_2$ fluxes in temperate salt marshes exhibit a
temperature dependency over seasonal scales, even in the presence of tides.

While $CO_2$ fluxes show seasonal patterns, there are no diel patterns that persist

throughout the year. During G1, the peak of high tide coincided with peak daily temperature.
This scenario also occurred during D1, but fluxes were too low to discern patterns. During all
other campaigns, low tide and peak temperatures coincided. These results suggest that diel
patterns may occur periodically under certain conditions. For example, at the SS site, it may be
that diel patterns occur during high tide at the temperature peak. While we expected the highest
fluxes during low tides due to increased oxygen exposure, there may be a lag between low tide in
the creek and low water levels at the SS site, resulting in higher fluxes during high tide in the
creek. However, these results can vary from site to site and with proximity to the tidal creek.
More research using high temporal frequency measurements are needed to parse out the role of
temperature and tides on $CO_2$ fluxes across salt marshes to properly represent the pattern in earth
system models (Ward et al., 2020)

Similarly to $CO_2$, $CH_4$ has a significant relationship with air temperature, however it

explains less variability in the fluxes. Several studies have found positive correlations between
soil $CH_4$ fluxes and temperature (Bartlett et al., 1985; Emery and Fulweiler, 2014; Wang and
Wang, 2017) in temperate salt marshes, while others have not (Wilson et al., 2015). It is
important to note that while, in general, salt marsh $CH_4$ fluxes are positively related to
temperature (Al-Haj and Fulweiler, 2020), the ability of temperature to explain $CH_4$ flux
variability is low, compounded by many, often site-specific, factors that affect methane



production and consumption, such as organic matter supply, microbial communities, and
diffusion rates (Al-Haj and Fulweiler, 2020; Bartlett et al., 1985).

At our study site, $CH_4$ fluxes were highest and pulses were most frequent during

senescence, agreeing with findings from ecosystem-scale measurements derived using the eddy
covariance technique (Vázquez-Lule and Vargas, 2021). In most wetland ecosystems, the highest
fluxes have been reported during the summer (Kim et al., 1998; Rinne et al., 2007; Van Der Nat
and Middelburg, 2000; Livesley and Andrusiak, 2012), but we highlight that there is a lack of
measurements during the winter (Al-Haj and Fulweiler, 2020). In *S. alterniflora* marshes, highest
mean $CH_4$ fluxes have been found in both the summer and the fall (Bartlett et al., 1985; Emery
and Fulweiler, 2014). At a site dominated by *S. alterniflora,* both high fluxes and porewater $CH_4$
concentrations were found in September, indicating either a continual build-up of $CH_4$ in the
pore water over the growing season and/or increased $CH_4$ production in the fall. For our site, it is
likely higher $CH_4$ emissions during senescence were due to an input of labile organic matter
from plant die-off (Seyfferth et al., 2020). Furthermore, a recent study has shown that porewater
DMS, a non-competitive substrate for methylotrophic methanogenesis that is produced from the
breakdown of DMSP, a metabolite produced by *S. alterniflora* (Dacey et al., 1987), peaks during
the fall (Tong et al., 2018). Therefore, we postulate that an influx of DMS may also contribute to
higher $CH_4$ fluxes during senescence in marshes dominated by *S. alterniflora*. This finding
highlights the importance of carbon-sulfur biogeochemistry and measuring fluxes during non-
summer months; particularly in marshes that have plant communities that provide substrates used
in methylotrophic methanogenesis (Seyfferth et al., 2020).

On a diel timescale, pulse emissions of $CH_4$ from the soil tend to occur during the

warmest time of the day, as well as during low and rising tides. There are very few studies that



report high-temporal frequency data of $CH_4$ emissions, most of which include plants within their
scope (via transparent chambers or eddy covariance) or focus on tidal creeks, making it difficult
to ascertain whether the diel patterns seen in this study are typical of tidal salt marsh soils.
Considering the broader range of studies about $CH_4$ fluxes in coastal vegetated ecosystems, $CH_4$
emissions have been found to peak at various points in the day, from during the day (Yang et al.,
2018, 2017; Tong et al., 2013), at night (Diefenderfer et al., 2018), or highly variable (Jha et al.,
2014; Xu et al., 2017). At our site, $CH_4$ fluxes tended to peak at the confluence of peak daily
temperature and low to rising tides, indicating that physical forcing may contribute to $CH_4$ pulses
(Bahlmann et al., 2015; Middelburg et al., 1996). However, pulses did occur during other times
throughout the day and within the tidal cycle. While some of the pulse emissions may be a result
of ebullition, the majority are associated with high $R^2$'s, indicating that they are sustained over
the measurement period. Our results demonstrate the importance of conducting high-temporal
frequency $CH_4$ measurements in tidal salt marsh soils for several reasons, including the need for
more data to better understand the drivers of $CH_4$ fluxes at diel scales and how that affects model
predictions.

$N_2O$ emissions and uptake loosely followed a seasonal pattern, likely driven by the

canopy phenological stages. During the growing season, it has been shown that highly
productive plants can compete with soil microbes for $NO_3^-$ and $NH_4^-$ (Cheng et al., 2007; Yu et
al., 2012; Zhang et al., 2013; Granville et al., 2021; Xu et al., 2017), shifting denitrifiers into
consuming $N_2O$ and resulting in a net uptake during G1 and M1. As the plants reach peak
maturity, the system shifts into net emission of $N_2O$ during M2 and S1. One study found that
nitrogen additions resulted in a pulse of $N_2O$ in July when most of the plant growth had occurred,
but no response in April, suggesting that the competition for $NO_3^-$ and $NH_3^+$ decreases when



plant growth has slowed down (Moseman-Valtierra et al., 2011). Increased substrate availability
combined with warm temperatures likely contributed to the marsh being a net source of $N_2O$
during the later stages of the growing season. As temperatures drop, the system shifts back into
net uptake, as seen during S2 and D1. Similar seasonal patterns have been seen in other studies,
albeit shifted by a month or two depending on the local climate and phenophases (Granville et
al., 2021; Emery and Fulweiler, 2014). These findings highlight balance between processes that
produce $N_2O$ (e.g., nitrification, denitrification, and nitrifier-denitrification) and consume $N_2O$
(e.g., denitrification), as well as substrate availability and plant phenology in determining
whether a marsh is a source or sink of $N_2O$ at any given point.

As with seasonality, diel patterns of $N_2O$ showed both emissions and uptake. Several

studies have also reported both emissions and uptake during a 24-hour period (Yang et al., 2017;
Tong et al., 2013). We found that pulses of uptake and emissions occurred both during the day
and at night, as well as during different phases of the tidal cycle. Studies have found higher
fluxes during the day (Tong et al., 2013; Yang et al., 2017) and at night (Laursen and Seitzinger,
2002; Yang et al., 2017; Bauza et al., 2002). Generally, fluxes were slightly higher at night
throughout the campaigns, perhaps as a result of increased availability of $NH_4^+$ at night due
decreased competition from photosynthesizers (Bauza et al., 2002). Overall, $N_2O$ fluxes were
near-zero with a $< 0.50$ nmol m$^{-2}$ s$^{-1}$ difference between daytime and nighttime mean fluxes,
suggesting that $N_2O$ fluxes do not play a major role in GHG emissions at this salt marsh.

Our automated measurements of sulfur-based trace gases show high variability in $CS_2$,

with low fluxes punctuated by occasional pulse emissions. There are no previous studies with
automated measurements to compare our findings, but previous studies have noted that $CS_2$
fluxes are highly variable (Steudler and Peterson, 1985; Hines, 1996), with periods of emission



and uptake. However, fluxes at SS were, on average, an order of magnitude higher than values
reported in the literature (Supplementary Table 1). There could be several reasons for the
difference in magnitudes: 1) improvement in instrumentation to detect $CS_2$, 2.) sampling
technique differences, and 3.) site-specific characteristics. Since the influx of sulfur-based trace
gas measurements in the 1980s, instrumentation has advanced from using molecular sieves and
cryotraps to store samples before measuring them on a gas chromatograph (e.g., Carroll et al.,
1986; Cooper et al., 1987; Steudler and Peterson, 1984) to using portable Fourier transform
infrared (FTIR) spectrometers that measure trace gas concentrations in near real-time. These
instrumentation advances subsequently led to changes in sampling techniques. Traditionally, it
was common to keep the chamber closed for upwards of 24-hours, with samples being collected
over hourly intervals throughout the day (Carroll et al., 1986; Goldan et al., 1987). Sweep air
free of sulfur trace gases was also commonly used to avoid the need to take samples at both the
inlet and outlets of the chambers (Goldan et al., 1987). However, others used ambient air because
it more closely resembled *in situ* conditions (Steudler and Peterson, 1985). With recent advances,
sampling techniques have changed to eliminate the need for very long closure times and reduce
the effects the chambers have on micrometeorological conditions. Now, high-temporal
frequency, long-term data can be obtained, thereby capturing pulse emissions that otherwise may
be missed. The third reason for difference in magnitude could be due to site-specific differences
in $CS_2$ fluxes. While the mechanisms by which $CS_2$ is produced are poorly understood, there are
several potential production pathways: OM degradation, photochemical production, and algal
production (Xie and Moore, 1999). The most likely pathway for our site is the microbially-
mediated reaction between $H_2S$ and organic matter due to high sulfur concentrations, anaerobic
conditions, and a large pool of decaying organic matter. Finally, $CS_2$ is a short-lived sulfur gas



but the major product of $CS_2$ oxidation is COS; consequently, understanding $CS_2$ production and
oxidation is important for recognizing the role of salt marshes in COS dynamics (Whelan et al.,

2013).

The mean of measured DMS fluxes generally fall within those reported in the literature,

but with pulses higher than previously reported and different temporal patterns. We found that
DMS fluxes only occurred during the middle of the day, near when air temperatures peaked. This
is contrary to several studies that have found DMS fluxes during other times of the day
(Morrison and Hines, 1990; Steudler and Peterson, 1985; DeLaune et al., 2002). Some studies
have found diel patterns related to temperature (De Mello et al., 1987; Cooper et al., 1987b) and
incoming tides (Morrison and Hines, 1990; Dacey et al., 1987; Goldan et al., 1987). Our results
indicate that DMS fluxes from the SS site are associated with temperature and light-related
processes, whether these variables influence microbial activity, plant physiology, or a
combination of both. A study found that DMS fluxes peaked after a full daylight period in a
Danish estuary (Jørgensen and Okholm-Hansen, 1985). However, there is no information on the
diel patterns of DMS in the sediment pore water or its release from *S. alterniflora* plants. DMS is
also produced by other pathways that occur under anoxic conditions, such as methylation of
sulfide and methanethiol (Lomans et al., 2002; Sela-Adler et al., 2015), microbial reduction of
dimethylsulfoxide (Capone and Kiene, 1988), and/or the incorporation of inorganic substrates
(i.e., $CO_2$) and organic methylated compounds (Finster et al., 1990; Moran et al., 2008; Lin et al.,
2010). To better understand DMS fluxes, more research into the dynamics between *S.*
*alterniflora*, pore water DMS, and DMS fluxes is needed, as it plays an important role in carbon-
sulfur biogeochemistry, particularly as a non-competitive substrate for methylotrophic
methanogenesis (Seyfferth et al., 2020) .




*4.2 Continuous versus discrete measurements: do we get the same information?*

Our results show that discrete temporal measurements of $CO_2$ during daytime low tide throughout the year (including dormancy) may be sufficient to obtain a representative mean of the temporal variability of soil $CO_2$ flux. This has implications for calculating carbon budgets. Furthermore, the distribution of continuous and discrete $CO_2$ fluxes is similar, indicating that discrete measurements are capturing similar variability as continuous measurements. This observation is reinforced by the $CO_2 \sim$ air temperature relationships, which do not have significantly different slopes (discrete: 0.03 - 0.12, continuous: 0.04 - 0.05), providing further support for the utility of daytime low tide discrete measurements in evaluating potential drivers of $CO_2$ variability.

In contrast, high variability in $CH_4$ fluxes resulted in the means for discrete and continuous measurements to be similar, but with significantly different distributions. In salt marshes, $CH_4$ fluxes are characterized by high variability (Rosentreter et al., 2021), making it difficult to assess the processes that control $CH_4$ fluxes (Vázquez-Lule and Vargas, 2021). While the means were not significantly different despite ~33% higher mean flux using discrete measurements, it is important to note that the 95% confidence interval and the coefficient of variation are broad and very high, resulting in potential error cancellation for the calculation of the mean. We postulate that the discrete measurement approach can be used to calculate budgets with the caveat of large uncertainties and that they likely overestimate the mean $CH_4$ flux. Discrete measurements do not capture similar variability as the continuous measurements and have a stronger air temperature-$CH_4$ flux relationship than continuous measurements, despite the overlap between their confidence intervals (2.14 - 12.7 and 1.31 - 2.71, respectively). However,





continuous measurements provide a more accurate depiction of the patterns and magnitudes of
$CH_4$ and can provide stronger insights into the interrelated drivers of $CH_4$ fluxes.

Regardless of the sampling interval, $N_2O$ fluxes had means that are near-zero. Due to

fluxes consistently being near zero, the discrete and continuous measurements will likely get
similar overall results due to error cancellation even if the distributions were significantly
different. The continuous measurements capture a wider range of fluxes than the discrete
measurements, as seen with its very high coefficient of variance and a different distribution.
However, the skewness between the two approaches is very similar, due to the bulk of the
measurements falling around the same values. It is important to note that this site is nitrogen-
limited, which constrains $N_2O$ production. In marshes that are not nitrogen-limited, sampling
intervals will likely play a more important role since fluxes will be higher.

For $CS_2$, discrete and continuous measurements did not have similar means or

distributions, likely due to the high variability found in these measurements. Previous studies
using discrete measurements of $CS_2$ have noted its high variability (e.g. De Mello et al., 1987),
with one highlighting the need for frequent measurements of sulfur-based trace gases during the
day in order to obtain an accurate mean daily flux value (Steudler and Peterson, 1985). We found
that discrete measurements taken during daytime low tide result in a daily mean that is nearly
twice that of the daily mean from the continuous measurements. The average $CS_2$ fluxes
measured during our field campaigns were up to an order of magnitude higher than previously
reported. We advocate for more measurements of $CS_2$ fluxes beyond focusing on low tide
windows and during different canopy phenological phases across salt marshes to better
understand the dynamics of this trace gas.




When measuring DMS fluxes during daytime low tide, the mean is similar to the

continuous measurement mean, but the distributions are significantly different. However, caution
should be taken in using discrete measurements of DMS to calculate daily means, particularly if
those measurements fall during the warmest part of the day when DMS fluxes are the most
active. This could result in overestimating the daily mean since extended periods of no fluxes are
not accounted for. One approach to measuring DMS fluxes would be to use the strong
relationship between discrete and continuous measurements to correct for the overestimation of
discrete fluxes. However, this approach would still require the use of a continuous, automated
system at different points throughout the year to establish a site-specific correction of discrete
mean DMS fluxes, particularly if DMS fluxes are used to calculate DMS budgets.

**5. Conclusion: what are we missing: potential caveats?**

Discrete measurements have the clear advantage of capturing the spatial variability of soil

trace gas fluxes across an ecosystem, but this approach is also used to describe the temporal
variability. Here we discuss the advantages and differences from discrete and continuous
measurements derived from this study. Discrete measurement campaigns are suitable for
calculating budgets, particularly for $CO_2$ and $N_2O$ since they capture very similar means. While
we found that $CH_4$ and DMS means were not significantly different between the two approaches,
there are caveats that must be considered when using discrete measurements. The high variability
inherent in $CH_4$ fluxes can contribute to the lack of significant differences between the two
approaches and result in discrete measurements overestimating the overall $CH_4$ fluxes from a
tidal salt marsh. This has implications when calculating SWGP where differences in $CH_4$ means
largely contribute to the differences in SGWP between the two approaches and can affect how



scientists and policymakers view tidal salt marshes and blue carbon as a natural climate solution
(Macreadie et al., 2021). For DMS, it is important to assess diel patterns to ensure that fluxes are
representative, particularly at sites that have patterns similar to what is seen at our study site.
When evaluating variability or trying to parse out the processes that drive GHG and trace gas
emissions from tidal salt marshes, using continuous, automated measurements would be the best
approach. This is particularly important for $CH_4$, where pulse emissions are frequent during the
growing season and can be very high. Using continuous measurements is also important in
scenarios where discrete measurements do not capture a similar mean or distribution, as with $CS_2$
fluxes. However, discrete measurements are more capable of representing spatial variability, and
until we have a better understanding of which source of variability is higher, temporal, or spatial,
both techniques should be considered for ecosystem assessments.

**Data availability**
Meteorological (station: delsjmet-p) and water quality (station: Aspen Landing) data are
available from the National Estuarine Research Reserve's Centralized Data Management Office
(CDMO) at https://cdmo.baruch.sc.edu/. Phenological data are available from the PhenoCam
network (site: stjones) at https://phenocam.sr.unh.edu/webcam/sites/stjones/. Data from trace gas
fluxes will be publicly available in a FAIR data repository (e.g., Figshare) before publication of
this research.

**Author contributions**
MC and RV conceptualized the study, designed the methodology, and conducted project
administration. MC conducted the formal analysis, investigation, and visualization, as well as



wrote the original draft. RV provide funding, resources, supervision, as well as reviewed and
edited the manuscript.

**Competing interests**

The authors declare that they have no conflict of interest.



**Acknowledgments**
This research was supported by the National Science Foundation (#1652594). MC acknowledges
support from an NSF Graduate Research Fellowship (#1247394). We thank the onsite support
from Kari St. Laurent and the Delaware National Estuarine Research Reserve (DNERR), as well
as from Victor and Evelyn Capooci for field assistance during the first campaign. We thank
George Luther for inspiring discussions about carbon-sulfur biogeochemistry in salt marshes.
The authors acknowledge the land on which they conducted this study is the traditional home of
the Lenni-Lenape tribal nation (Delaware nation).



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
