# Peer review of "Trace gas fluxes from tidal salt marsh soils: implications for carbon- # 2 sulfur biogeochemistry"

_Biogeosciences, 2022_

## Author Response (AR1)

**Associate Editor**
**Comments to the author**:
Both reviewers were very positive about your manuscript, commenting on both the quality of the writing and interesting and pioneering science Therefore, I am recommending minor revisions with an editorial review afterwards. Please make sure to respond to the revisions suggested by both reviewers.

Response: We appreciate the support for this manuscript and the recognition of the novelty of this work. We have addressed all the minor revisions to improve clarity of this manuscript. Furthermore, we have uploaded the final dataset in Figshare (Data from trace gas fluxes are available in Figshare (doi:10.6084/m9.figshare.20449131; see lines 637-638).

**RC1 comments**
Comment: In general, the study presents useful contribution in our understanding of trace gas fluxes from saltmarsh ecosystem. I believe that manuscript is well-written and fits well in the frame of the basic requirement for Biogeosciences Discussion.

Response: We thank the reviewer for their kind comments and support on the manuscript.

Comment: Please find below some specific recommendations which may be useful.
Introduction: This section covers the background information nicely.

Response: Thank you!

Comment: Line 65: I suggest finding some recent references.

Response: We added the following reference on L65-66 from 2014:
Brimblecombe, P.: The Global Sulfur Cycle, in: Treatise on Geochemistry, vol. 10, edited by:
        Holland, H.D. and Turekian, K.K., Elsevier Science, 559-591,
        http://dx.doi.org/10.1016/B978-0-08-095975-7.00814-7, 2014

Comment: Lines 92-98: Please provide references.

Response: We added examples of automated measurements in salt marshes on L100-101:
Diefenderfer, H.L., Cullinan, V.I., Borde, A.B., Gunn, C.M., Thom, R.M.: High-frequency
        greenhouse gas flux measurement system detects winter storm surge effects on salt
        marsh, Glob. Chang. Biol., 24, 5961-5971, http://doi.wiley.com/10.1111/gcb.14430,
        2018.
Capooci, M., Vargas, R.: Diel and seasonal patterns of soil $CO_2$ efflux in a temperate tidal marsh,
        Sci. Total Environ., 802, https://doi.org/10.1016/j.scitotenv.2021.149715, 2022.

The challenges in installing instruments in tidal salt marshes listed in lines 100-103 are based on our own experiences installing this type of equipment in tidal salt marshes as there have been very limited published examples.

Methods:
Comment: Line 134: How the position of the collars was selected to best represent the study area? What was the extent of the sampled area/total area?

Response: The location of the collars was defined by several constraints. One, our power source for the autochambers and the greenhouse gas analyzers was line power via outlets interspersed throughout an access point in a boardwalk. Therefore, we were constrained to locations ~15 m away from the outlets due to the length of the tubing connecting the autochambers to the

analyzers. This is a known limitation using the LI-8100 multiplexer and most of any autochambers using fixed tubing lengths for their analyzers. Two, St. Jones Reserve is a protected wetland and we are required to have permits approved by the State of Delaware in the USA. Therefore, we designed this experiment to minimize impact to the ecosystem and follow approved guidelines of the State of Delaware. These state guidelines constrained our experiment to areas near the available outlet and the access point in a boardwalk. Details of collar placement are now provided in L143-145.

      With that being said, collars are located in a short *Spartina* area, which comprises of ~66% of the vegetated cover in the marsh (Vázquez-Lule and Vargas, 2021; as mentioned in line 132). Each collar covered an area of ~314 cm$^2$ for a total of ~1,884 cm$^2$. While we acknowledge that fluxes are heterogenous across the landscape, average $CO_2$ fluxes throughout the campaigns were similar to those measured in prior studies covering a larger spatial extent of this wetland (L355-357; Capooci and Vargas, 2022; Seyfferth et al., 2020; Hill et al 2022).

References
Hill, A. C., and R. Vargas. 2022. Methane and carbon dioxide fluxes in a temperate tidal salt marsh: Comparisons between plot and ecosystem measurements. Journal of geophysical research. Biogeosciences 127. http://doi.org/10.1029/2022JG006943
Capooci, M. and Vargas, R.: Diel and seasonal patterns of soil CO2 efflux in a temperate tidal marsh, Sci. Total Environ., 802, https://doi.org/10.1016/j.scitotenv.2021.149715, 2022.
Seyfferth, A. L., Bothfeld, F., Vargas, R., Stuckey, J. W., Wang, J., Kearns, K., Michael, H. A., Guimond, J., Yu, X., and Sparks, D. L.: Spatial and temporal heterogeneity of geochemical controls on carbon cycling in a tidal salt marsh, Geochim. Cosmochim. Acta, 282, 1–18, https://doi.org/10.1016/j.gca.2020.05.013, 2020.
Vázquez-Lule, A. and Vargas, R.: Biophysical drivers of net ecosystem and methane exchange across phenological phases in a tidal salt marsh, Agric. For. Meteorol., 300, https://doi.org/10.1016/j.agrformet.2020.108309, 2021.

Comment: Lines 135-137: Please describe how the disturbance to the soil was addressed while removing vegetation? Was all vegetation removed?

Response: Vegetation was removed by carefully clipping the base of the stem where it met the soil surface. All vegetation within each collar was carefully clipped prior to each campaign. We confirmed that vegetation clipping had minimal to no impact on fluxes, as seen with a lack of anomalous fluxes during the beginning of each campaign (Fig. 2). We clarified the fact that vegetation was clipped in L146-147.

Comment: Line 192-195: It seems to me that you have subsamples within the same area that presents the risk of pseudo replication. I suggest providing some more details about that.
Response: We clarified that the experimental design was limited by the technique of how automated chambers work. Arguably, manual soil flux measurements (i.e., with survey chambers) can cover a larger area but have limited replication in time (e.g., Vargas et al 2011). Automated chambers have the advantage of improving our information of temporal variability, but they are limited in spatial coverage. This technique is limited by how long the tubing

connecting a multiplexer can extend to where an autochamber is located (usually ~15-20 m). Having multiple systems to connect autochambers in distant places across a wetland has very high costs and will require providing electrical power in different ways. Consequently, most studies using automated chambers only use one multiplexer with multiple autochambers extending to a limited area (Barba et al 2018). That said, our results show the large spatial and temporal variability of these soil fluxes regardless of the spatial constraint of the autochambers. We will add a brief discussion on this topic in a revised version of the manuscript.

References

Barba, J., A. Cueva, M. Bahn, G. A. Barron-Gafford, B. Bond-Lamberty, P. J. Hanson, A. Jaimes, L. Kulmala, J. Pumpanen, R. L. Scott, G. Wohlfahrt, and R. Vargas. 2018. Comparing ecosystem and soil respiration: Review and key challenges of tower-based and soil measurements. Agricultural and Forest Meteorology 249:434–443.

Vargas, R., M. S. Carbone, M. Reichstein, and D. D. Baldocchi. 2011. Frontiers and challenges in soil respiration research: from measurements to model-data integration. Biogeochemistry 102:1–13.

Results

Comment: Line 334: I could not see Table 2 mentioned in the text

Response: Thank you for finding this mistake. We will include a reference to Table 2 in line 337.

Discussions:

Comment: Line 380-383: Is this generalisation for temperate wetlands?

Response: This pattern has been reported in northern, temperate, and subtropical wetlands by Turetsky et al. (2014) in a synthesis of $CH_4$ emissions from 71 wetlands. We included this reference in L411, as well as added clarification in L408-409.

Reference

Turetsky, M.R., Kotowska, A., Bubier, J., Dise, N.B., Crill, P., Hornibrook, E.R.C., Minkkinen, K., Moore, T.R., Myers-Smith, I.H., Nykänen, H., Olefeldt, D., Rinne, J., Saarino, S., Shurpali, N., Tuittila, E-S., Waddington, J.M., White, J.R., Wickland, K.P., Wilmking, M.: A synthesis of methane emissions from 71 northern, temperate, and subtropical wetlands. Glob. Chang. Biol., 20, 2183-2197, https://doi.org/10.1111/gcb.12580, 2014.

Comment: Line 385-387: Reference needed

Response: The following reference is included in L415.

Zhang, Y., Ding, W.: Diel methane emissions in stands of Spartina alterniflora and Suaeda salsa from a coastal salt marsh. Aquat. Bot., 95, 262-267, https://doi.org/10.1016/j.aquabot.2011.08.005, 2011.

Comment: Line 401: Reference needed

Response: The references for the studies that have high-frequency data, but include plants within their scope are included in the following sentences on L432-436 in the discussion of the various diel patterns that have been found in coasted vegetated ecosystems. We rephrased L432-433.

Comment: Conclusion: Well done. Very interesting read.

Response: Thank you very much!

**RC2 comments**
Comment: This is a pioneering study that compares traditional discrete low-tide gas ($CO_2$, $CH_4$, $N_2O$, $CS_2$, DMS) flux measurements from a salt marsh against continuous, high-temporal frequency measurements. The novel approach to measuring high-frequency, continuous (72-hour) fluxes of gases is capable of capturing episodic, pulse emissions that might otherwise evade discrete, low tide measurements. High-temporal flux measurements of $CO_2$ did not differ from discrete measurements demonstrating that daily mean low tide measurements capture the variability of this process sufficiently, whereas the episodic spikes of the other gas fluxes ($CH_4$, $N_2O$, $CS_2$, DMS) that were captured by high-temporal measurements were missed by discrete measurements. There was also a strong relationship between temperature and $CO_2$ and $CH_4$ fluxes. However, precautions should be taken when using daily mean low tide (discrete) values to calculate annual flux budgets or warming potentials due to differences in flux ranges despite the mean values between the two approaches generally agreeing. Overall, this study is very thorough and the sound results are presented in very clear writing.

Response: We thank the reviewer for their kind comments and support for the manuscript and the study design.

Comment: Below are comments/questions that may improve the clarity of the manuscript text and figures, along with requests for more methodological details.
L45: add "can potentially" after also. Carbon storage rates have a vast range and many factors will impact the long-term storage potential.

Response: We included "can potentially" in L45.

Comment: L65-72: please specify the affect these compounds have on the climate, i.e., cooling or warming effects

Response: DMS has a cooling effect on the climate. $CS_2$ has a warming effect, but it is a precursor to carbonyl sulfide which is a cooling gas. $CS_2$ has a short lifetime (~days). We will include this within lines 65-72, as well as the following references. Text regarding DMS has been included in L67-69, while text regarding $CS_2$ has been included in L76-79.
References

Brühl, C., Lelieveld, J., Crutzen, P.J., Tost, H.: The role of carbonyl sulphide as a source of stratospheric sulphate aerosol and its impact on climate. Atmos. Chem. Phys., 12, 1239-1253, https://doi.org/10.5194/acp-12-1239-2012, 2012.
Charlson, R.J., Lovelock, J.E., Andreae, M.O., Warren, S.G.: Oceanic phytoplankton, atmospheric sulphur, cloud albedo and climate. Nature, 95(16), 655-661, https://doi.org/10.1038/326655a0, 1987
Thomas, M.A., Suntharalingam, P., Pozzoli, L., Rast, S., Devasthale, A., Kloster, S., Feichter, J., Lenton, T.M.: Quantification of DMS aerosol-cloud-climate interactions using the ECHAM5-HAMMOZ model in a current climate scenario. Atmos. Chem. Phys., 10, 7425-7438, https://doi.org/ 10.5194/acp-10-7425-2010, 2010.

Watts, S. F.: The mass budgets of carbonyl sulfide, dimethyl sulfide, carbon disulfide and hydrogen sulfide. Atmos. Environ., 34, 761–779, https://doi.org/10.1016/S1352-2310(99)00342-8, 2000.

Comment: L82-87: it is my understanding that disentangling gas fluxes during high tide are difficult since the effluxing gases are mixing/dissolving into the flood tide waters. Perhaps this should also be mentioned here. How were gas fluxes measured when the collars were underwater from the flood tide?

Response: The following sentences were included in L88-91: "Measurements at high tide in salt marshes are difficult due to both reduced access to the marsh platform and reduced fluxes. Gases from the soil mix with the overlying water column and move more slowly through water compared to air, contributing to a decline in fluxes during high tide."

With regards to flood tide at our site: The SS site rarely floods during high tide due to its distance from the tidal creek as well as the presence of a berm located adjacent to the tidal creek. As a result, the SS site floods during spring high tides and storm events. If the soils were flooded during the study period, it was minimal and measurements continued to run. To clarify the SS site's hydrology, details and references will be included in Section 2.1, L133-135.

References

Moffett, K.B., Wolf, A., Berry, J.A., Gorelick, S.M.: Salt marsh-atmosphere exchange of energy, water vapor, and carbon dioxide: Effects of tidal flooding and biophysical controls. Water Resour. Res., 46(W10525), https://doi.org/ 10.1029/2009WR009041, 2010.
Seyfferth, A. L., Bothfeld, F., Vargas, R., Stuckey, J. W., Wang, J., Kearns, K., Michael, H. A., Guimond, J., Yu, X., and Sparks, D. L.: Spatial and temporal heterogeneity of geochemical controls on carbon cycling in a tidal salt marsh, Geochim. Cosmochim. Acta, 282, 1–18, https://doi.org/10.1016/j.gca.2020.05.013, 2020.

Comment: L128-133: this text reveals that the gas fluxes are measured over ~four days; this makes the use of the word "continuous" in the introduction misleading. I understand that continuous over four days is still different than discrete low tide measurements, but this should be clarified in the Introduction that this study utilizes a continuous, multi-day measurements. Perhaps it's my own bias, but to me continuous implies year-round.

Response: We agree that the use of "continuous" can imply different things. Here "continuous" denotes that the measurements were performed in an automatic and continuous way until the instruments were shut down for each campaign. We clarified in the introduction (L110-111) that continuous measurements lasted ~72 hours. At this moment it is not possible to perform long-term continuous measurements because of electrical power limitations and damage to the instruments due to the high salinity content in the environment. We discussed this in more detail in L161-162.

Comment: L135-137: could benthic microalgae be present on the sediment surface? Were gas chambers darkened to prevent photosynthesis? Were plant stems trimmed down to the sediment surface or pulled out completely? Could remaining plant structure in the sediment act as "straws" or conduits of gas exchange? If plants were completely removed (i.e., pulled out or cut to below the surface), was the sediment surface disturbed?

Response: Here are the answers to the above five questions:
1. We did not see the presence of dense microbial mats on the sediment surface. However, microalgae not forming mats could have been present but were not evident with a "naked eye".
2. Chambers were opaque. We clarified that in L149.
3. Plant stems were clipped to the sediment surface. We clarified that in L148-149.
4. It is entirely possible that the plant structure in the sediment could act as "straws" for transport of gases to the atmosphere. However, plant stems were trimmed on a frequent enough basis, such that stem diameters were reduced, minimizing the effects of plant-mediated transport of trace gases. That said, we discussed this mechanism in L149-150.
5. Plants were not completely removed to minimize disturbance to the sediment surface. These plants have dense thick rhizomes and removing them will represent a major disturbance to the soil structure. Therefore, careful clipping was the most effective approach. Finally, our measurements of $CO_2$ and $CH_4$ fluxes were comparable to manual measurements performed with chambers of different sizes and including different components of vegetation in this wetland (Hill et al 2022; L359-362)

Reference
Hill, A. C., and R. Vargas. 2022. Methane and carbon dioxide fluxes in a temperate tidal salt marsh: Comparisons between plot and ecosystem measurements. Journal of geophysical research. Biogeosciences 127. http://doi.org/10.1029/2022JG006943

Comment: L140: what was the volume of a chamber? Did this warrant an internal fan to homogenize gases while measuring?

Response: The chamber volume was 4071.1 cm$^3$, added . LICOR soil chambers do not have fans included in their design. As per the website (https://www.licor.com/env/products/soil_flux): "mixing is achieved through a bowl-shaped chamber and air inlet/outlet positioning." Therefore, the chambers were designed to homogenize gases during chamber closure. These are "industrial type" autochambers and have been extensively engineered to address changes in pressure and to maximize flow for measurements.

Comment: L155-158: The description of the QAQC is a bit too brief. There seems to be a lot of steps packed into this sentence. It would help the reader to have each step in a sentence, at least, with more description. I recognize that every step of QAQC cannot be divulged in detail, but the current state of this sentence is far from reproducible.

Response: We have followed previously published QA/QC protocols but lines 171-177 were rewritten as follows: QAQC included several steps. First, all values due to instrumental errors

such as an insufficient chamber closure seal were removed. These errors were identified by the SoilFluxPro software. Second, the $R^2$ for the linear and exponential fits of trace gas emissions were compared and the fit with the higher $R^2$ was chosen. Third, all fluxes that occurred when the $R^2$ of $CO_2$ was <0.90 were removed. Low $R^2$'s indicate that the soil micrometeorological conditions were not stable during the measurement. Finally, all negative $CO_2$ fluxes were removed since they were likely erroneous".

Comment: L216: salinity should be reported as unitless

Response: Salinity is a measure of the amount of salt per unit volume (e.g., gr/L) and it is usually expressed as ppt (part per thousand).

Comment: Figure 2: the text reporting mean, UCI, and LCI is too small. Please enlarge. Define UCI and LCI in the figure caption.

Response: We thank the reviewer for catching that we forgot to define the UCI and LCI in the figure caption. We will included the following definitions: UCI = upper 95% confidence interval, LCI = lower 95% confidence interval on L286. We enlarged the mean, LCI, and UCI in the figure as best we can.

Comment: Figure 4: is it possible to produce higher resolution figures so the density curves do not look pixelated? The grey color is not included in the legend or described in the caption. It needs to be. I assume this is where overlap occurs. If so, it appears the N2O fluxes are completely and perfectly overlapped, which was not mentioned in the text.

Response: We noticed that the R code for the graph produces pixelated density curves. We tried alternative methods of plotting the density curves, but we ran into the same pixelation issue in all versions, so the figure remains the same. We included the color of the overlap between continuous and discrete measurements in the legend. $N_2O$ fluxes had similar means for both the continuous and discrete measurements. We added a reference to the density curves on L550-551.

Comment: Figure 5: if I understand this correctly, this plot compares the continuous measurement (over 72 hours) to the discrete measurement (one hour before and after low tide). If so, the time frames should be mentioned in the figure caption and/or text to remind the reader how exactly the comparison is made.

Response: The description by the reviewer is correct. We added the study's definition of continuous and discrete in the appropriate legends (i.e., Fig. 4 (L336-337), 5 (L348-349), and 6 (L355-356), Table 1(L340-341) and 2 (L362-363)).

Comment: L366-368: it may be useful to mention the findings of McTigue et al. 2021 (doi: 10.3389/fmars.2021.661442) that demonstrate the relationship between CO2 production and temperature is a function of the activation energy required to breakdown salt marsh sediment organic matter.

Response: We included the reference in the preceding paragraph (L380-382) which discusses the role of temperature in $CO_2$ fluxes.

Comment: L387: "pore water" should be one word, as is used throughout the rest of the manuscript

Response: Thank you for catching that – it has been corrected.

Comment: L568: remove the comma after "temporal"

Response: Thank you for catching that – the comma has been removed.